# AdpA Positively Regulates Morphological Differentiation and Chloramphenicol Biosynthesis in *Streptomyces venezuelae*

Małgorzata Płachetka,[a] Michał Krawiec,[a] Jolanta Zakrzewska-Czerwińska,[a] Marcin Wolański[a]

[a]Faculty of Biotechnology, University of Wrocław, Wrocław, Poland

**ABSTRACT** In members of genus *Streptomyces*, AdpA is a master transcriptional regulator that controls the expression of hundreds of genes involved in morphological differentiation, secondary metabolite biosynthesis, chromosome replication, etc. However, the function of AdpASv, an AdpA ortholog of *Streptomyces venezuelae*, is unknown. This bacterial species is a natural producer of chloramphenicol and has recently become a model organism for studies on *Streptomyces*. Here, we demonstrate that AdpASv is essential for differentiation and antibiotic biosynthesis in *S. venezuelae* and provide evidence suggesting that AdpASv positively regulates its own gene expression. We speculate that the different modes of AdpA-dependent transcriptional autoregulation observed in *S. venezuelae* and other *Streptomyces* species reflect the arrangement of AdpA binding sites in relation to the transcription start site. Lastly, we present preliminary data suggesting that AdpA may undergo a proteolytic processing and we speculate that this may potentially constitute a novel regulatory mechanism controlling cellular abundance of AdpA in *Streptomyces*.

**IMPORTANCE** *Streptomyces* are well-known producers of valuable secondary metabolites which include a large variety of antibiotics and important model organisms for developmental studies in multicellular bacteria. The conserved transcriptional regulator AdpA of *Streptomyces* exerts a pleiotropic effect on cellular processes, including the morphological differentiation and biosynthesis of secondary metabolites. Despite extensive studies, the function of AdpA in these processes remains elusive. This work provides insights into the role of a yet unstudied AdpA ortholog of *Streptomyces venezuelae*, now considered a novel model organism. We found that AdpA plays essential role in morphological differentiation and biosynthesis of chloramphenicol, a broad-spectrum antibiotic. We also propose that AdpA may undergo a proteolytic processing that presumably constitutes a novel mechanism regulating cellular abundance of this master regulator.

**KEYWORDS** *Streptomyces*, master transcriptional factors, regulation of gene expression, secondary metabolite gene clusters

**M**any free-living bacteria exhibit a complex life cycle that includes cell differentiation (e.g., spore formation) and/or the development of multicellular structures (e.g., hyphae, biofilms). These processes are often triggered by environmental stresses, such as nutrient depletion, physical/chemical factors, and signaling molecules. To cope with this and coordinate the response with cell cycle progression, bacteria utilize complex regulatory networks of dedicated transcriptional regulators. Some of these regulators also control other key cellular processes, such as chromosome replication and/or secondary metabolite production, and thereby exert pleiotropic effects on cell functioning. These proteins are often referred to as global regulators. The most extensively studied examples include the Spo0A, CtrA, MrpC, and AdpA proteins of genera *Bacillus*, *Caulobacter*, *Myxococcus*, and *Streptomyces*, respectively. Various mechanisms strictly regulate the intracellular abundance and activities of these regulators at different

Address correspondence to Marcin Wolański, marcin.wolanski@uwr.edu.pl.

stages of gene expression to ensure the coordinated progression of development during growth (1–6) (for reviews see also references 7–9).

AdpA orthologs are ubiquitously present in the members of genus *Streptomyces* and play essential roles in their morphological differentiation (10–16). Most of the AdpA orthologs positively regulate morphogenesis, with the exception of that recently reported in *Streptomyces xiamenensis* (17). Colonies of *Streptomyces* mutants devoid of AdpA are characterized by their inability to produce fluffy aerial hyphae on solid media; such mutants were historically described as "bald" (18, 19) and are unable to produce spores. In addition to their roles in development, AdpAs are involved in the biosynthesis of many secondary metabolites, including antibiotics (10, 11, 13, 20–24). A study in the model species, *Streptomyces griseus*, showed that AdpA can bind ~1,500 sites on the chromosome and directly regulate (mostly activate) the expression of several hundred genes. The AdpA regulon includes many genes involved in morphological differentiation and primary and secondary metabolism (e.g., pyrimidine and streptomycin biosynthesis, respectively), as well as its own gene (25). AdpA was also found to be involved in regulating chromosome replication (26). The rather degenerate sequence of the AdpA consensus binding sequence (5′-TGGCSNGWWY-3′, referred to as the AdpA box) is reflected by its low DNA-binding specificity. This feature of AdpA is assumed to facilitate the generation of complexes involving other transcription factors, thereby helping generate regulatory networks to mediate the transcriptional control of many genes (25).

To ensure the coordination of morphological differentiation and secondary metabolite biosynthesis, the bacterial cell must precisely control the level of AdpA. In *Streptomyces*, *adpA* gene expression is subject to complex regulation by multiple mechanisms at different levels. First, the transcription factors ArfA, BldD, ArpA, and SlbR have been identified to bind the *adpA* promoter region and directly regulate *adpA* gene transcription in various *Streptomyces* species (11, 27–29). The activities of some of these proteins are modulated by specialized signaling metabolites, such as hormone-like butyrolactones (SlbR and ArpA) and cyclic-di-GMP (BldD) (28, 30, 31). The *adpA* gene is also subject to autoregulation. This was found to be controlled by different modes in the model species, *S. griseus* and *S. coelicolor*: in the former, AdpA negatively affects the transcription of its own gene, whereas in the latter, AdpA acts as an activator (32, 33). A possible explanation for this discrepancy is proposed later in this text. At the posttranscriptional level, a mechanism of targeted *adpA* mRNA degradation by AbsB/RNase III was identified in *S. coelicolor* (34). The wide conservation of AbsB in various streptomycetes (35) suggests that posttranscriptional processing of the *adpA* mRNA could constitute a universal mechanism in *Streptomyces*. AdpAs are believed to be targets for translational control due to the presence of a rare leucine TTA codon in the coding sequences of their genes (12, 14, 36, 37). However, a few *adpA* genes of streptomycetes lack this TTA codon (38). So far, there has been no report of a mechanism that directly regulates the level or DNA-binding activity of the AdpA protein.

*Streptomyces venezuelae* is a recently established model for developmental and secondary metabolite studies (39, 40). In comparison to the model streptomycetes, *S. griseus* and *S. coelicolor*, *S. venezuelae* exhibits faster growth and can differentiate to completion in liquid medium, which facilitates the application of various omics techniques and time-lapse microscopy in *Streptomyces* research (41–47). Although a number of reports have examined developmental regulation in *S. venezuelae* (30, 48–52), little is known about the role of the master regulator, AdpA, in this model. *S. venezuelae* is known as a producer of secondary metabolites, including antibiotics such as chloramphenicol, jadomycin B, and pikromycin (53–61). The genomes of the various *S. venezuelae* strains carry an average of ~30 biosynthetic gene clusters for secondary metabolites (62, 63). However, as seen for many other *Streptomyces*, most of the putative secondary metabolites of *S. venezuelae* have never been detected under laboratory conditions because there is little or no transcription of the corresponding gene clusters (64–66). Activating the expression of these gene clusters is considered crucial for the discovery of novel secondary metabolites. However, little is known about the

complexity of the relevant regulatory pathways, making it difficult for researchers to activate secondary metabolite expression despite the use of molecular engineering tools and/or heterologous host expression. Several studies have examined the biosynthesis of chloramphenicol by its natural producer, *S. venezuelae*, leading to the identification of a few regulators (CmlR, MtrA, JadR1) involved in the transcriptional regulation of the corresponding gene cluster (53, 67, 68). The regulator, CmlR (also referred to as vnz_04400 or sven0913), is present in the chloramphenicol gene cluster and was found to play a crucial role in activating gene expression within this cluster. A very recently proposed mechanism explains the interplay between CmlR and the global transcriptional repressor, Lsr2, in regulating gene expression within the chloramphenicol biosynthetic gene cluster (Cm-BGC) (69). Despite these recent discoveries, however, we still know little about the roles of MtrA, JadR1, and AdpA in *S. venezuelae*.

Here, we aimed to characterize the function of AdpASv, an AdpA counterpart encoded in *S. venezuelae*, in the development and gene autoregulation of this new model species. Our results suggest a possible new mechanism that may regulate AdpA protein abundance in *Streptomyces* cells and highlight the importance of AdpA in chloramphenicol biosynthesis. Finally, we compare the functions of AdpASv with those of the AdpA orthologs in the "older," more extensively studied models, *S. coelicolor* and *S. griseus*.

## RESULTS

**Disruption of the *adpA$_{Sv}$* gene affects *S. venezuelae* differentiation.** The AdpA of *S. venezuelae*, AdpASv, shares ~83% to 90% overall amino acid sequence identity with its orthologs from other *Streptomyces* species (Fig. 1A and Fig. S1). Notably, AdpASv exhibits 100% sequence identity with the other species within its C-terminally located DNA-binding domain, which contains two helix-turn-helix (HTH) motifs responsible for DNA sequence recognition and DNA binding, and very high sequence identity (92 to 99%) within its N-terminal oligomerization domain. The C-terminal portion of AdpASv beginning at 325 amino acids (aa), directly following the DNA-binding domain, comprises the least conserved part of this protein, showing up to 54% identity with SGR_4742 and SCLAV_1957 of *S. griseus* and *S. clavuligerus*, respectively, and no significant similarity with other counterparts. A significant level of variability in this region is observed in all analyzed AdpA orthologs. Our phylogenetic analysis of protein sequences revealed that, among the tested proteins, AdpASv is related more closely to its ortholog from *S. griseus* (AdpASg) than to that from *S. coelicolor* (AdpASc) (Fig. S2A). The relationships on amino acid levels reflect phylogenetic relationships between *Streptomyces* species based on classical phylogenetic marker 16S rRNA gene sequences (Fig. S2B). On the gene organization level, the *adpA$_{Sv}$* (vnz_12630) locus largely resembles the *adpA* loci of other *Streptomyces* species (Fig. S3). The upstream *uspA* and downstream *ornA* genes encoding universal shock protein and RNase, respectively, are conserved at these locations in all analyzed *Streptomyces*. In *S. venezuelae* and a few other *Streptomyces*, however, the directly adjacent region upstream of *adpA* carries a convergently orientated hypothetical gene that is not present in the genomes of the "older" model organisms, *S. coelicolor* and *S. griseus*.

To examine whether AdpA affects morphological differentiation in *S. venezuelae*, as it does in other *Streptomyces*, we first constructed an *adpA* disruption mutant strain (Sven_ΔadpA) and a complemented strain harboring *adpA$_{Sv}$* under the native promoter at the *attP$_{\varphi BT1}$* integration site (Sven_ΔadpA/adpA-FLAG; also used for Western blotting). We assessed the phenotypes of these strains on various solid media commonly used for characterizing *Streptomyces* species: maltose-yeast extract-malt extract (MYM), soya flour mannitol (SFM), International Streptomyces Project yeast malt extract agar medium (ISP-2), and modified DSMZ medium 65 (GYM). On the most common media, MYM and SFM, which are used to facilitate the sporulation of many other *Streptomyces* species, we observed the production of aerial hyphae by the wild-type (Sven_WT) and complemented Sven_ΔadpA/adpA-FLAG strains but not by Sven_ΔadpA,

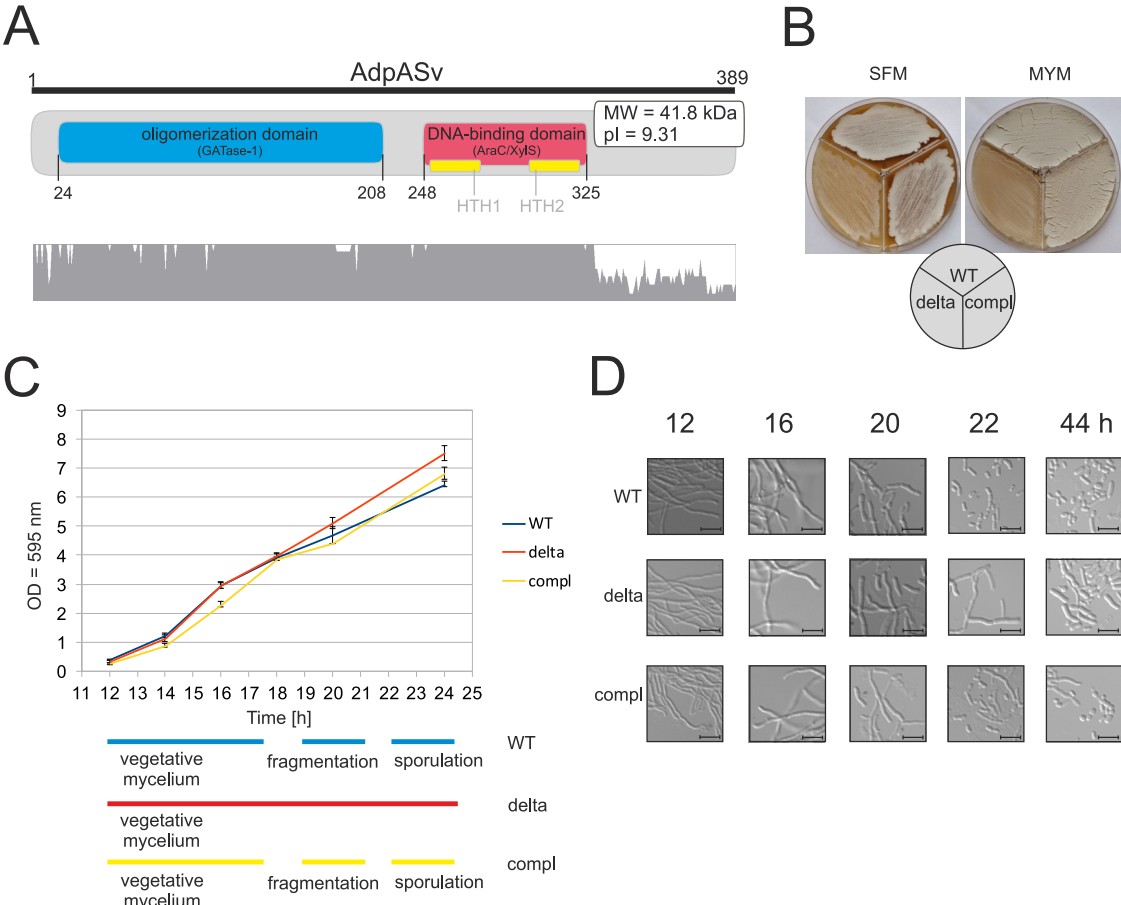

**FIG 1** *In silico* analysis of AdpASv and phenotypic characteristic of *S. venezuelae* mutants. (A) Domain organization of AdpASv based on NCBI BLASTp search for conserved domains (upper panel). Amino acid sequence conservation of AdpA orthologs in seven *Streptomyces* species (bottom panel; for details see Fig. S1). The graph was created using the Ugene software (ClustalW algorithm). (B) Phenotypes of the wild-type Sven_WT (WT), *adpA* disruption mutant Sven_ΔadpA (delta), and complemented mutant Sven_ΔadpA/adpA-FLAG (compl) strains grown for 6 days on solid SFM and MYM media. (C) Growth curves of *S. venezuelae* strains cultivated in liquid MYM medium (strain designations are as listed in panel B). Relative developmental schemes based on microscopic differential interference contrast (DIC) images are shown below (based on panel D). (D) Microscopic DIC images of corresponding *S. venezuelae* cultures in liquid MYM medium (representative images). Scale bar (black bar) represents 5 μm.

which exhibited the characteristic bald phenotype (Fig. 1B). None of the tested *S. venezuelae* strains (with the surprising exception of the Sven_ΔadpA mutant on GYM) was able to differentiate on the other two (relatively similar) media, ISP-2 and GYM (Fig. S4A). This difference may reflect the presence of calcium ions in ISP-2 medium but not GYM medium (see Discussion).

In addition, by placing *adpA$_{Sv}$* under the control of the inducible *tcp830* promoter, we examined how elevation of AdpA expression influenced the phenotypes of the *S. venezuelae* strains. We introduced the pAVadpA plasmid (*adpA$_{Sv}$* under the control of the *tcp830* promoter) into the wild-type and *adpA* disruption mutant strains (Sven_WT/adpA$^+$ and Sven_ΔadpA/adpA$^+$, respectively) and compared their growth on media containing different concentrations of the inducer, anhydrotetracycline (ATET). In the Sven_ΔadpA/adpA$^+$ mutant, complementation of the bald phenotype was observed only after the medium was supplemented with ATET, thus confirming production of AdpA (Fig. S4B). Upon induction with ATET, only negligible differences in colony growth rates and phenotypes were observed between the wild-type strain containing an additional inducible copy of *adpA$_{Sv}$* (Sven_WT/adpA$^+$) and the control wild-type strain (Sven_WT), suggesting that an increased AdpA level does not affect morphogenesis (Fig. S4B). Since *S. venezuelae* can undergo its complete differentiation cycle in liquid MYM medium, we assessed how *adpA*

affects growth and differentiation of *S. venezuelae* in this milieu. Our time course analysis showed that all three strains (wild-type, disruption mutant, and the complemented mutant) exhibited similar growth rates (Fig. 1C). To assess differentiation, we checked the ability of the deletion mutant to generate spores. The spore suspension took on the intense greenish color indicative of abundant spore production for Sven_WT but not for the Sven_ΔadpA mutant strain (Fig. S4C). This is in line with our microscopic observations. The *adpA* deletion mutant exhibited only occasional spores at 44 h of growth, whereas both the wild-type and the complemented mutant strains exhibited extensive sporulation as early as 22 h of growth (Fig. 1D). In addition to hampered spore production, cultures of Sven_ΔadpA were lighter in color than those of the wild-type strain, likely indicating the absence of one or more brown-pigmented secondary metabolite (Fig. S4D).

Finally, we examined the gene expression and protein levels of AdpA during *S. venezuelae* development. For this purpose, samples collected from different time points during growth in liquid MYM medium were subjected to reverse transcriptase quantitative PCR (RT-qPCR) and Western blotting. RT-qPCR showed that the $adpA_{Sv}$ transcription profiles were similar between the wild-type Sven_WT and complemented Sven_ΔadpA/adpA-FLAG strains: their $adpA_{Sv}$ mRNA levels were relatively stable throughout the assessment period (12 to 24 h), with the exception of the 14 h time point, when a few times higher signals were observed for both analyzed strains (Fig. 2A). The relative abundance of AdpA was examined using anti-FLAG antibody against cell extracts of the complemented Sven_ΔadpA/adpA-FLAG strain producing C-terminally FLAG-tagged AdpASv. We observed that the signal from AdpA-3×FLAG fluctuated during the assessment period: it was notably high at the earliest time point (12 h, vegetative growth) and clearly detectable between 18 and 22 h (onset of sporulation) but undetectable between 12 and 18 h (late vegetative stage) and after 22 h (sporulation stage) (Fig. 2B). Notably, the AdpA-3×FLAG protein profile dropped quickly from its highest observed level observed at 12 h to below the limit of detection within only 2 h.

Collectively, these results show the following. AdpASv shares high amino acid sequence identity with other AdpA orthologs and full sequence conservation within the DNA-binding domain of the protein. AdpA is essential for the morphological differentiation of *S. venezuelae* on both solid and liquid media. Transcription of $adpA_{Sv}$ exhibits a relatively stable profile during the growth cycle with a pronounced peak at 14 h, while AdpA protein abundance experiences significant fluctuations during cultivation. Both the $adpA_{Sv}$ transcript and the AdpA protein are most abundant at early stages of growth before the start of aerial hyphae formation.

**AdpASv positively regulates the expression of its own gene.** To assess whether AdpASv is similar to its counterpart AdpAs from model *Streptomyces* in the ability to autoregulate $adpA_{Sv}$ gene transcription, we assessed its capability to interact with and affect the activity of its own gene promoter region.

First, we analyzed *in silico* the presence and organization of AdpA binding sites within the *adpA* promoter regions of *S. venezuelae*, *S. coelicolor*, and *S. griseus* ($padpA_{Sv}$, $padpA_{Sc}$, and $padpA_{Sg}$, respectively). As these AdpA proteins have identical DNA-binding domains, we performed our binding-site search using the well-established AdpASg consensus binding sequence, 5'-TGGCSNGWWY-3' (AdpA box) (70). We identified a single strong AdpA box (one mismatch) in the $adpA_{Sv}$ promoter region and at least one perfect (no mismatch) and two strong AdpA boxes for $padpA_{Sc}$ and $padpA_{Sg}$ (Fig. S5A). Notably, the strong box identified in $padpA_{Sv}$ was present in all three examined *adpA* promoters (gray box in Fig. S5A).

Next, we examined AdpA-*padpA* interactions *in vitro* with an electrophoretic mobility shift assay (EMSA). For this assay, we used recombinant His-tagged AdpA proteins of *S. venezuelae* and *S. coelicolor* (AdpASv_His6 and AdpASc_His6, respectively) (Fig. S6A and B) and Cy5 fluorescently labeled DNA fragments corresponding to the promoter regions of $adpA_{Sv}$, $adpA_{Sc}$, and $adpA_{Sg}$. EMSA analysis revealed that both recombinant proteins specifically interacted with all tested *adpA* promoter regions and

A

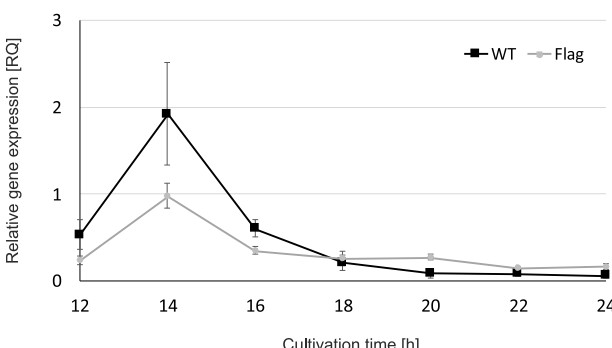

B

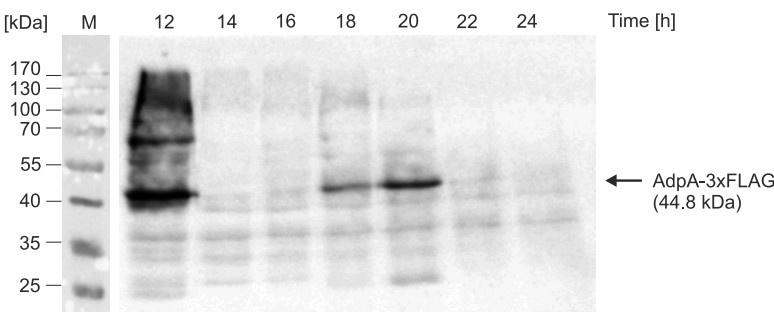

**FIG 2** Gene expression of *adpA*~Sv~. (A) Transcriptional analysis of *adpA*~Sv~ in Sven_WT and Sven_ΔadpA/ adpA-FLAG strains. Samples were collected at 2-h intervals from three independent cultures of *S. venezuelae* grown in liquid medium and assessed using RT-qPCR. Relative gene expression (RQ) levels were calculated with normalization to the level of the sigma factor homolog, *hrdB*; *S. venezuelae* chromosome (RQ = 1) was used as a calibrator. (B) Western blotting of AdpA expression in Sven_ΔadpA/adpA-FLAG. Cell lysates were prepared from liquid cultures collected at the indicated time points, and their equal amounts (40 $\mu$g per lane) were loaded on the gel (see protein loading control in Fig. S6C). Upon transfer, the AdpA-3×FLAG fusion protein was specifically detected using an anti-FLAG antibody conjugated with horseradish peroxidase (see no-FLAG control, Fig. S6D). M, protein mass marker.

showed less specific interaction with the negative-control fragment (NC, lacking an AdpA box) (Fig. 3A). Given the high sequence homology between AdpA proteins, especially within the DNA-binding domain, we expected that they would show similar mode of DNA bindings. Indeed, the AdpASv_His6 and AdpASc_His6 proteins exhibited similar binding modes toward the analyzed DNA fragments, which was reflected by nearly identical patterns of nucleoprotein complexes.

The interaction of AdpASv with its own promoter was further confirmed *in vivo* by chromatin immunoprecipitation combined with qPCR (ChIP-qPCR). For this purpose, we analyzed the complemented mutant strain Sven_ΔadpA/adpA-FLAG at 12 and 20 h of growth in liquid medium, representing time points at which we observed AdpA production (Fig. 2B). Indeed, samples derived from the FLAG-tagged AdpA-producing strain were significantly enriched for the DNA fragment harboring the *padpA*~Sv~ region at both time points (Fig. 3B). The enrichment for *padpA*~Sv~ expressed as relative quantity (RQ) was 36.7 and 15.3 for samples collected at 12 h and 20 h of growth, respectively. Conversely, we did not observe enrichment of the *padpA*~Sv~ fragment in a control sample (Sven_WT strain at 20 h), indicating that the eluted DNA bound specifically to AdpA protein, not the affinity resin. No substantial difference in enrichment levels between the samples was observed for another tested fragment, *oriC* region (serving

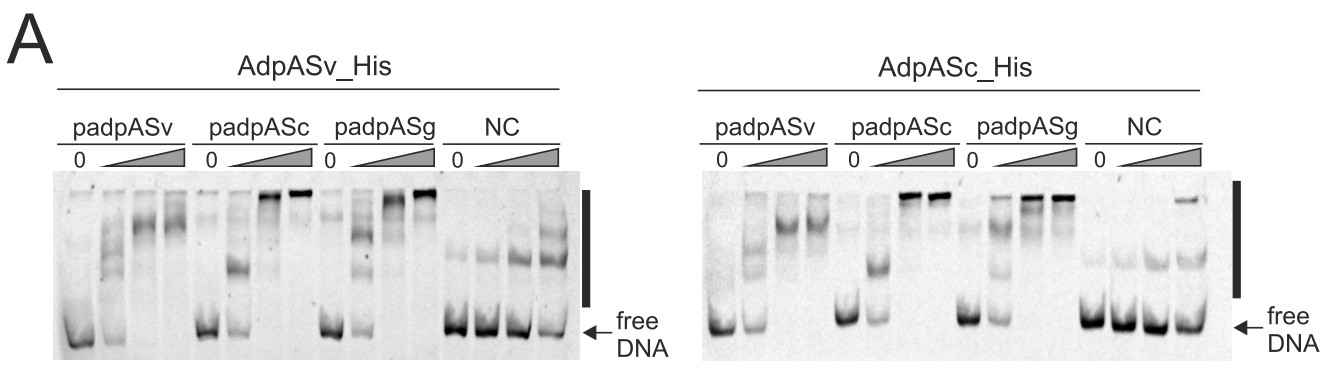

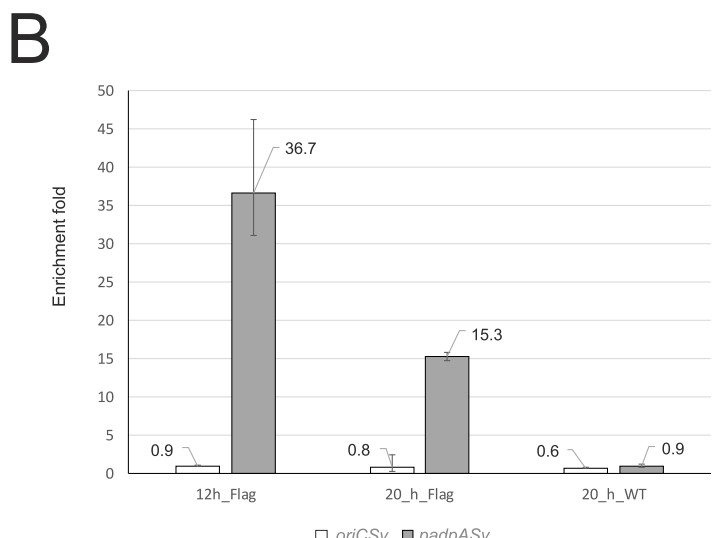

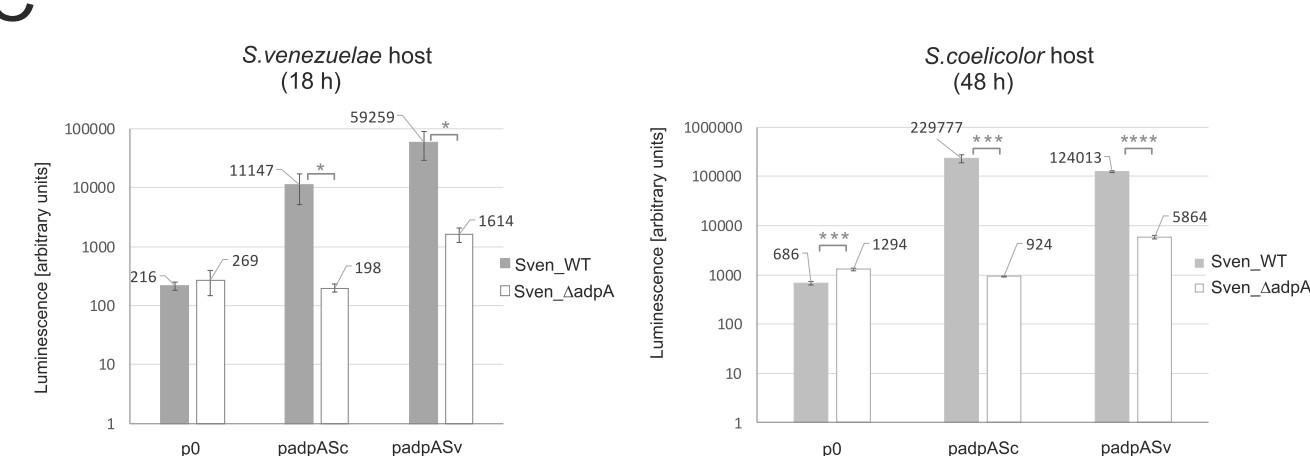

**FIG 3** *In vitro* and *in vivo* interactions of AdpA proteins with their targets. (A) EMSAs of the interactions of AdpASv and AdpASc with *adpA* promoter regions of various *Streptomyces* species. Cy5 dye-labeled DNA fragments encompassing the *adpA* promoters of *S. venezuelae*, *S. coelicolor*, and *S. griseus* (padpASv, padpASc, and padpASg fragments, respectively) were incubated with increasing concentrations (10, 50, and 250 nM, gray triangles) of the respective recombinant AdpA proteins or without AdpA (lane 0). NC, negative-control DNA fragment (HP0500) lacking the AdpASg consensus binding sequence. Nucleoprotein complexes are collectively marked by vertical solid lines along both gels. (B) ChIP-qPCR analysis of the binding of AdpASv-3×FLAG to the *padpASv* and *oriCSv* regions in liquid cultures of Sven_ΔadpA/adpA-FLAG strain. The levels of DNA fragments recovered from ChIP were assessed using qPCR, and relative quantification was performed; the results are presented as the fold-enrichment. A genomic region lacking an AdpA binding sequence and a sample derived from Sven_WT (12 h) lacking FLAG-tagged AdpA served as the normalizer and calibrator, respectively. ChIPs were performed at 12 and 20 h of growth. The graphs represent the averages of technical triplicates for two biological replicates. (C) Analysis of $adpA_{Sv}$ and

as a negative control, see below), indicating specific interaction of AdpA protein with *adpA* promoter.

Next, we investigated the ability of AdpASv to transcriptionally regulate its own gene. To do this, we constructed an integrative luciferase reporter pFLUXH vector harboring the *luxCDABE* genes under the control of *padpA$_{Sv}$* (pFLUXH-padpA$_{Sv}$) and introduced it into the wild-type and Δ*adpA S. venezuelae* strains. In addition, to examine the binding of AdpASv and AdpASc proteins to heterologous *adpA$_{Sc}$* and *adpA$_{Sv}$* promoters, we constructed pFLUXH-padpA$_{Sc}$ harboring the *adpA$_{Sc}$* promoter and generated the respective heterologous *S. venezuelae* and *S. coelicolor* host strains (Table S1). The data obtained from luminescence assays revealed that the *adpA$_{Sv}$* promoter was several dozen times less active in the native and heterologous host strains lacking *adpA* than in the corresponding wild-type strains (Fig. 3C, both panels; Fig. S7A and B). A similar dependency was observed for the *adpA$_{Sc}$* promoter in both hosts. The *adpA$_{Sv}$* and *adpA$_{Sc}$* promoters demonstrated higher luminescent activities in the presence of their cognate genes in the native host than in the heterologous host. Using the same method, we also sought to examine the activity of the *adpA* promoter of *S. griseus* (*padpA$_{Sg}$*). However, the *padpA$_{Sg}$* activities measured in the wild-type and *adpA* deletion mutant strains of *S. coelicolor* were, for some unknown reason, highly irreproducible (data not shown). Despite several attempts, we were unable to obtain *S. venezuelae* clones containing a pFLUXH-padpA$_{Sg}$ harboring *padpA$_{Sg}$*.

Collectively, these results show that AdpASv, similar to AdpASc, positively autoregulates the activity of the *adpA$_{Sv}$* gene promoter by directly binding to its own gene promoter region. Moreover, both AdpASv and AdpASc interchangeably activate the promoters of orthologous *adpA* genes, although they activate their cognate gene promoters more effectively.

**AdpASv does not interact with *oriCSv*.** Previous studies revealed that AdpA can bind to the origin of chromosome replication (*oriC*) in *S. coelicolor* and *S. griseus* and may be involved in regulating chromosome replication at the initiation stage of *S. coelicolor* (25, 26).

Similar to our analysis of *adpA* promoters, we analyzed the presence and organization of AdpA boxes within *oriC* sequences of the model species, *S. venezuelae*, *S. coelicolor*, and *S. griseus* (*oriC$_{Sv}$*, *oriC$_{Sc}$*, and *oriC$_{Sg}$*, respectively). These searches revealed that *oriC$_{Sv}$* contains four AdpA boxes, one perfect and three strong, whereas *oriC$_{Sc}$* and *oriC$_{Sg}$* contain three and two strong AdpA boxes, respectively (Fig. S5B). In *S. coelicolor* and *S. griseus*, all identified AdpA boxes are located in the 5′ parts of the *oriC* regions. In the case of *S. venezuelae*, only one AdpA box (the perfect one) is located in the 5′ portion of *oriC$_{Sv}$*; the remaining strong AdpA boxes are positioned in the central part of *oriC$_{Sv}$*.

To verify the possible binding of AdpASv to its native origin of replication region, *oriC$_{Sv}$*, we applied the above-described ChIP-qPCR approach. However, in those experiments, we failed to observe enrichment of the DNA fragment harboring the *oriC$_{Sv}$* region (Fig. 3B). Consistently, we could not identify *oriC$_{Sv}$* among the fragments identified by preliminary analyses of our ChIP-seq data (data not shown). We therefore did not perform any further *in vitro* investigation of a putative interaction between recombinant AdpASv and the *oriC$_{Sv}$* region.

In sum, these data suggest that AdpA of *S. venezuelae*, in contrast to its counterparts in *S. coelicolor* and *S. griseus*, does not interact with its native *oriC$_{Sv}$* region *in vivo* and thus probably does not regulate chromosome replication at the initiation stage.

**FIG 3** Legend (Continued)
*adpA$_{Sc}$* gene promoter activities using luciferase reporter assays in *S. venezuelae* and *S. coelicolor* hosts. Strains harboring pFLUXH vectors containing *adpA* gene promoter regions of *S. coelicolor* or *S. venezuelae* (padpASc or padpASv, respectively) or control empty plasmid (p0) were grown on solid medium and measured at the indicated time points. Each bar represents the average of three medians (derived from 4 technical replicates) obtained for three independent clones; the error bars represent standard deviations; statistical significances are indicated (*, $P < 0.05$; ***, $P < 0.001$; ****, $P < 0.0001$). Graphs were generated in Excel.

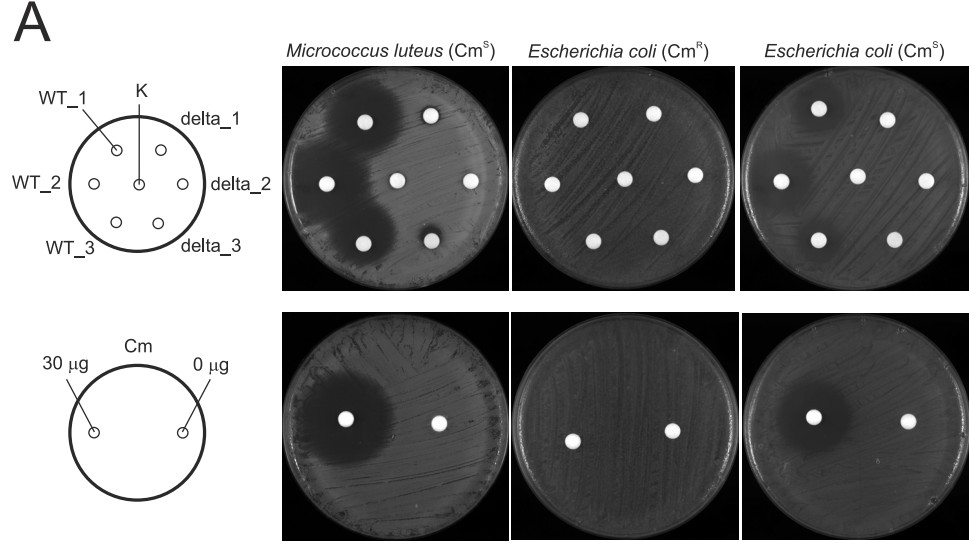

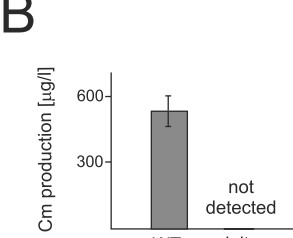

**FIG 4** Chloramphenicol production in *S. venezuelae*. (A) Antibacterial activities of culture broth extracts derived from *S. venezuelae* strains, as assessed using a disk diffusion method. Extracts obtained from three independent cultures (designations 1 to 3) of Sven_WT (WT) and Sven_ΔadpA (delta) strains were tested against the chloramphenicol-sensitive indicator strain, $Cm^s$ (*Micrococcus luteus* and *Escherichia coli* DH5$\alpha$), and chloramphenicol-resistant indicator strain, $Cm^R$ (*E. coli* ET12567) (upper panel). The same indicator strains were tested against defined amounts of chloramphenicol (bottom panel). The control disks contained either the solvent (K) or 30 $\mu$g of antibiotic in solvent solution. (B) Analysis of chloramphenicol production titer in liquid cultures of *S. venezuelae*. The assay was performed by disk diffusion method using *M. luteus* as an indicator strain and the disk containing either Sven_WT strain culture extracts or defined amounts of chloramphenicol (see Fig. S8). The graph represents the average titer of chloramphenicol (Cm) production calculated from three independent cultures and the standard deviation. ND, antibiotic activity not detected. Graph was prepared in Excel.

**AdpASv is essential for chloramphenicol production.** In *S. coelicolor* and *S. griseus*, AdpAs have been shown to control the production of secondary metabolites, such as actinorhodin and undecylprodigiosin in *S. coelicolor* and streptomycin in *S. griseus*.

Our *in silico* search of the *S. venezuelae* genome sequence using the AdpASg consensus sequence, 5′-TGGCSNGWWY-3′, allowed us to identify putative AdpA binding sites in the promoter regions of several genes within the chloramphenicol biosynthetic gene cluster (Cm-BGC) (Table S3 and Fig. S9). To investigate the possible role of AdpA in regulating this gene cluster, we used the modified Kirby-Bauer disk diffusion method to compare the production of chloramphenicol in the wild-type and *adpA* deletion mutant strains. We prepared liquid culture extracts of Sven_WT and Sven_ΔadpA strains grown in GYM production medium, which was previously applied to study chloramphenicol biosynthesis (53), and tested them against chloramphenicol-sensitive *Micrococcus luteus* and *Escherichia coli* DH5$\alpha$ strains. To confirm that the observed effect was due solely to the activity of chloramphenicol, we used chloramphenicol-resistant *E. coli* ET12567 strain harboring the chromosomally integrated chloramphenicol acetyltransferase (*cat*) gene. The chloramphenicol acetyltransferase is responsible for attaching an acetyl group to chloramphenicol, thus preventing chloramphenicol from binding to ribosomes. We observed that the extracts obtained from Sven_WT strain inhibited the growth of both chloramphenicol-sensitive strains but not that of *E. coli* ET12567 (Fig. 4A, upper

panels). In contrast, the extracts derived from Sven_ΔadpA cultures did not produce growth inhibition zones against either of the tested strains. As expected, the complemented *adpA* deletion mutant Sven_ΔadpA/adpA-FLAG exhibited restored antimicrobial activity against *M. luteus* (Fig. S8A). Consistently, in control experiments including disks soaked with 30 μg of chloramphenicol standard, the *M. luteus* and *E. coli* DH5α strains showed large growth inhibition zones while the growth of *E. coli* ET12567 was not inhibited (Fig. 4A, bottom panels). The results suggest that the growth inhibition effect observed for culture extracts derived from Sven_WT was due mainly to the presence of chloramphenicol. To quantify chloramphenicol production in the wild-type strain, we again applied the disk diffusion method. Based on a standard curve representing the relationship between the size of the growth inhibition zone and the antibiotic concentration (Fig. S8B to D), we estimated the chloramphenicol production titer of Sven_WT to be 550 μg/L, whereas no production was detectable for the Sven_ΔadpA mutant (Fig. 4B).

In sum, these results indicate that $adpA_{Sv}$ is essential for chloramphenicol biosynthesis in *S. venezuelae* and further suggest that AdpASv may directly regulate the transcription of genes within the chloramphenicol biosynthetic gene cluster.

## DISCUSSION

The members of genus *Streptomyces* undergo a complex life cycle that includes morphological differentiation of the hyphae. Numerous signals and transcription factors are involved in orchestrating development in these bacteria (reviewed in reference 71). Among them, the pleiotropic regulator, AdpA, has been studied extensively over the past 2 decades due to its importance in morphological development, the production of secondary metabolites (including antibiotics), and the regulation of chromosome replication (for reviews see references 36, 40, 72, 73). However, no previous study has examined the function of an AdpA ortholog in *S. venezuelae*, which was recently established as a model for developmental studies in *Streptomyces*. Here, we report for the first time that AdpA plays essential roles for differentiation and chloramphenicol biosynthesis in its native strain producer, *S. venezuelae*.

AdpAs are ubiquitously present in *Streptomyces* (38). AdpASv exhibits high degrees of similarity to other AdpA counterparts at both the amino acid and domain organization levels (Fig. 1 and Fig. S1). Moreover, the local synteny of the $adpA_{Sv}$ locus region largely resembles those from other analyzed *Streptomyces* genomes (Fig. S3). Notably, however, the regions upstream of *adpA* differ between *S. venezuelae* and the older model organisms, *S. coelicolor* and *S. griseus*, due to the presence of an additional gene of unknown function (vnz_12625). Phylogenetic analysis reflects similar relationships between AdpA and 16S rRNA sequences in the examined *Streptomyces* species (Fig. S2). This suggests that AdpA homologs presumably coevolved from a common ancestor and likely retain similar biological functions, as suggested previously (38). Our phenotypic observations made on common solid cultivation media, like SFM and MYM, provide evidence that, similar to the AdpAs in most other *Streptomyces* (10–14, 16, 74), AdpASv is required for aerial hyphae development and consequently for spore formation (Fig. 1 and Fig. S4). The wild-type strain used in this study produced the green spores typical for *S. venezuelae* NRRL B-65442 strain when grown on MYM medium (75). Interestingly, we observed striking phenotypic differences between the wild-type and *adpA* deletion mutant strains when grown on the *Streptomyces* media, IWSP-2 and GYM (Fig. S4): wild-type Sven_WT was not able to produce aerial hyphae on either medium, whereas deletion mutant strain Sven_ΔadpA produced fluffy hyphal structures on GYM (but not ISP-2) medium. The only difference between the ISP-2 and GYM is the presence of calcium ions in the latter. We thus speculate that this phenotypic difference could be attributed to the presence of calcium ions. Indeed, previous studies showed that various metal ions, including calcium, could influence the growth and spore formation of *Streptomyces* (76–79).

A number of previous studies showed that *adpA* gene expression can be regulated at the transcriptional, posttranscriptional, and translational (TTA codon) levels (11, 12,

14, 34). However, none of the previous reports described any posttranslational regulation of AdpA protein. In our study, we observed pronounced alterations in AdpASv levels during the cell cycle (Fig. 2). The highest AdpA abundance (12 h) preceded the peak of $adpA_{Sv}$ mRNA transcript (14 h), suggesting that there may be positive autoregulation of $adpA_{Sv}$ gene expression; indeed, the results of our luciferase reporter assay were consistent with this notion (Fig. 3C). The immediate disappearance of AdpA from cell lysates at 14 h may indicate that AdpA undergoes efficient proteolysis. Another discrepancy between the transcript and protein levels was noted for the 18 to 22 h time span, when the $adpA_{Sv}$ transcript level was rather stable while the protein level exhibited a significant increase potentially due to repression of proteolysis. Thus, we hypothesize that these fluctuations in AdpA protein levels indicate the existence of a mechanism for targeted AdpA proteolysis. Consistent with this notion, our preliminary *in vitro* analyses revealed specific proteolysis of purified recombinant N-terminally FLAG-tagged AdpASv incubated with a cell lysate of wild-type *S. venezuelae* cell lysate (Western blotting, unpublished data). Interestingly, similar posttranslational processing of AdpA may take place in *S. griseus*; in this case, the *in vivo* proteolysis of AdpASg was observed for N-terminally tagged His-AdpA but not for the C-terminally labeled protein, suggesting that protein proteolysis occurred specifically at the C terminus (25). Thus, we propose that the degradation of AdpA in *S. griseus* and *S. venezuelae* may take place at the C-terminal portion of the protein, which encompasses the DNA-binding domain. We speculate that the proteolytic targeting of this part of AdpA could abolish its interaction with DNA and thereby serve as a mechanism to regulate the expression of AdpA-dependent genes, including its own gene.

In *S. venezuelae* liquid cultures, the repressed level of AdpA returned to a relatively high level during the hyphae fragmentation stage (18 to 22 h), suggesting that it is important for this differentiation. Our laboratory is currently using an RNA-seq approach to investigate the role of AdpASv in global gene expression during different stages of growth in *S. venezuelae*.

At the transcriptional level, *adpA* expression is subject to autoregulatory control, as demonstrated in several previous reports (17, 23, 32, 33). However, the autoregulatory effect that AdpA exerts on its own gene expression seems to differ across *Streptomyces* species. In *S. griseus*, the best studied AdpA model, AdpA appears to negatively affect the transcription of its own gene, whereas in *S. coelicolor* and *S. lincolnensis*, it functions as an activator. In the present work, we demonstrate that AdpA of *S. venezuelae* exerts a positive impact on $adpA_{Sv}$ gene expression (Fig. 3C). Sequence alignment and *in silico* searches within *adpA* promoter regions indicated that the three examined model *Streptomyces* species each have unique arrangements of AdpA boxes (Fig. S5A). They shared the presence of one strong AdpA box (one mismatch) located in similar positions relative to the gene start codon, but this strong AdpA box was positioned differently in relation to the transcriptional start sites (TSSs) of the *adpA* genes in these species (discussed below). Previous transcriptional analyses demonstrated that the number and locations of *adpA* gene TSSs differ across these species (11, 14) (the TSS recognition for $adpA_{Sv}$ was based on unpublished data provided by personal communication from M. Bush and M. Buttner; see also the supplemental material). Of the analyzed species, the AdpA strong box lies downstream of the TSS only in *S. griseus*. This suggests that the binding of AdpA in this location can hinder the binding of RNA polymerase and thus inhibit *adpA* gene transcription, as proposed previously (32). In contrast, the binding of AdpA to sites located upstream of the TSS in the *adpA* promoter presumably facilitates the recruitment of RNA polymerase in a manner similar to that seen for other genes at which AdpA acts as an activator (32, 70). Our analyses indicate that in both *S. venezuelae* and *S. coelicolor*, the AdpAs positively regulate the transcription of their own genes (Fig. 3C); importantly, they also interchangeably bind and activate the counterpart *adpA* promoters of the other species, although higher activities were observed for the cognate promoter-protein pairs than for the heterologous

pairs (Fig. 3A and C). This cross-species AdpA-*adpA* promoter reactivity can be explained by the high similarity within the C-terminal portion encoding the DNA-binding domain and a similar arrangement of AdpA boxes in relation to the TSS. Our EMSAs further showed that the two AdpA orthologs bound the individual promoters in a similar way (Fig. 3A). Unfortunately, we were not able to include the promoter of $adpA_{Sg}$ in this comparative study, as we encountered problems with generating reporter strains in *S. venezuelae* and low reproducibility of measurements in *S. coelicolor*. We speculate that this could at least partially reflect that AdpA acts on $padpA_{Sg}$ via a different mode of regulation and/or an incompatibility/lack of other regulators in the heterologous host. Together, our results prompt us to hypothesize that the arrangement of AdpA boxes in relation to the TSS is likely to predominantly determine the mode of autoregulatory action exerted by an AdpA protein on its cognate *adpA* promoter. It should be mentioned that, in addition to autoregulation, *adpA* expression can also be controlled at the transcriptional level by a number of other proteins in different *Streptomyces*, e.g., ArfA, BldD, ArpA, and SlbR (11, 27–29). Thus, it seems that *adpA* promoters are adjusted to their hosts to enable the coordinated action of numerous regulators, including AdpA's own protein, to modulate *adpA* gene expression.

A previous study revealed that AdpA can also play a regulatory role in the chromosome replication of *S. coelicolor* (26). In the present study, our *in silico* analysis identified perfect and strong AdpA boxes within the origin of replication of *S. venezuelae* ($oriC_{Sv}$). However, we did not observe binding of AdpASv to $oriC_{Sv}$ in ChIP experiments (Fig. 3B), and an *in vitro* approach using EMSA showed that the AdpASv-$oriC_{Sv}$ interaction is rather weak (data not shown). *In silico* analyses indicated that the organization of putative AdpA boxes within the *oriC* regions of all three species differs significantly even though these *oriCs* share a relatively high level of sequence similarity (above 77% identity) and nearly identical arrangement of the binding sites for the replication initiator, DnaA (Fig. S5B and reference 80). In *S. coelicolor*, AdpA competes with DnaA for binding to *oriC* and thus presumably hinders DnaA-dependent DNA unwinding. Thus, we assume that the relative arrangement of AdpA and DnaA boxes must play a crucial role in the functioning of both proteins. As our *in vivo* experiments were conducted in liquid cultures for *S. venezuelae* and on solid media for *S. coelicolor*, we cannot exclude the possibility that AdpASv interacts with $oriC_{Sv}$ under other environmental conditions. However, we did not further study this aspect.

Several previous reports indicated that AdpA contributes to the biosynthesis of various secondary metabolites in *Streptomyces* (10, 11, 13, 20–23). Here, we demonstrate for the first time that *adpA* of *S. venezuelae* might be essential for the production of chloramphenicol, since disruption of this gene abolishes the biosynthesis of this medically important antibiotic. Four other regulators responsible for controlling gene expression within Cm-BGC have been identified to date, including both global regulators (MtrA and Lsr2) and local regulators (JadR1 and CmlR, which are involved in the production of jadomycin and chloramphenicol, respectively) (65–69). Binding sites within the chloramphenicol gene cluster were previously identified for these regulators but not for AdpA. In this study, we used an *in silico* approach to identify five putative AdpA binding sites within the promoters of genes comprising the Cm-BGC, namely, vnz_04400 (also referred to as *cmlR*), which is involved in the regulation of gene expression, vnz_04410 and vnz_04415 (also referred to as *cmlN* and *cmlF*, respectively), involved in chloramphenicol transport, and vnz_04455 and vnz_04460 (also referred to as *cmlI* and *cmlM*, respectively), involved in chloramphenicol biosynthesis (Fig. S9 and Table S3). The former gene encodes CmlR, which is a key transcriptional regulator that is indispensable for chloramphenicol synthesis (53, 69). Since some of these genes form operons (Fig. S9), our analysis suggests that AdpA might control at least several genes within Cm-BGC. Interestingly, our preliminary analysis of ChIP-seq results (data not shown) indicates that AdpASv binds *cmlR* upstream region, suggesting that AdpA controls biosynthetic genes via regulating *cmlR* transcription. However, it is also

possible that AdpA could control other promoters within Cm-BGC in different cultivation conditions, as seen for other regulators (e.g., JadR1). We are currently investigating regulation of gene expression in Cm-BGC in our laboratory.

The regulation of gene transcription in the Cm-BGC is still not clear. Recent discoveries indicate that the master regulator of secondary metabolism, Lsr2, may repress gene expression within Cm-BGC by looping distant regions of this gene cluster (69) and that CmlR, by binding within *cmlI-cmlM* intergenic region, can counteract Lsr2-mediated repression to allow transcription of biosynthetic genes. The putative binding sites of AdpA and CmlR overlap within the *cmlI-cmlM* region, suggesting that there may be an interplay between these two proteins. The *cmlN-cmlF* and *cmlI-cmlM* intergenic regions, which are bound by the regulators MtrA and JadR1, respectively, were also herein found to contain putative AdpA binding sites (Fig. S9), leading us to suspect that this regulatory network might be even more complex than previously thought. We speculate that, in addition to the above-mentioned regulators, other factors (e.g., cultivation conditions and possibly novel regulators) may affect the binding of AdpA within Cm-GBC. Additional studies are warranted to clarify the role of AdpA in regulating Cm-BGC. It is obvious that the overall mechanism of gene expression in the chloramphenicol gene cluster is complex and still remains elusive.

In sum, AdpA is a conserved master regulator that exerts pleiotropic and diverse effects on *Streptomyces* biology. Numerous studies, including ours, indicate that the unique arrangement of AdpA binding sites determines the binding targets for the AdpA protein and consequently the processes controlled by AdpA. This enables AdpA to control various metabolic pathways in different *Streptomyces* species. In the three model *Streptomyces* species studied herein, AdpA is required for both morphological differentiation and the production of secondary metabolites. In the present work, we report the first functional analysis of an AdpA ortholog of *S. venezuelae* and describe its roles in morphological differentiation, the autoregulation of its own gene transcription, and chloramphenicol biosynthesis.

## MATERIALS AND METHODS

**Bacterial strains and culture conditions.** The strains used in this work are listed in Table S1 of the supplemental material. *Escherichia coli* strains were grown at 37°C (or different temperatures as indicated) using solid or liquid Luria-Bertani medium (81) supplemented with antibiotics if necessary. *Streptomyces venezuelae* strains were grown at 30°C on solid media: MYM (maltose-yeast extract-malt extract) (49, 82), SFM (soya flour mannitol) (83, 84), ISP-2 (International *Streptomyces* Project yeast malt extract agar medium) (85), modified DSMZ medium 65 (hereafter, GYM: malt extract 10 g, yeast extract 4 g, glucose 4 g, CaCl$_2$ 1.46 g [pH 7.3], agar 16 g; ingredients per 1 L) (82, 84, 86), or liquid GYM medium (GYM without agar). Liquid cultures were conducted in 250-mL flasks (with or without baffles, as indicated) containing 50 mL of medium inoculated with *S. venezuelae* spores to a final optical density OD (600 nm) of ~0.001 to 0.002 (as indicated) with shaking (220 rpm).

**DNA manipulation and plasmid/strain construction.** The plasmids used in this study are listed in Table S1 of the supplemental material. DNA manipulations in *E. coli* were carried out using standard protocols (81). Reagents and enzymes were supplied by Thermo Fisher Scientific and Sigma-Aldrich. Oligonucleotides were synthesized by Sigma-Aldrich (Table S2). *S. venezuelae* mutants were constructed using a PCR-targeting strategy, according to the method described previously for *Streptomyces coelicolor* (84). For details on plasmid and strain construction, please see the supplemental material.

**Recombinant protein expression and purification.** For details on the protein purification protocol, please see the supplemental material. Briefly, the AdpA proteins of *S. venezuelae* and *S. coelicolor* were purified as recombinant C-terminally His-tagged AdpASv (3×Flag-AdpA-6×His; also called AdpASv_His) and C-terminally His-tagged AdpASc (AdpASc_His) (33), respectively, using the *E. coli* Rosetta 2(DE3) (Merck) strain harboring the pET28a-3×FLAG-adpA_Sven plasmid (for plasmid construction details, see the supplemental material). Each protein was isolated from a 1.6-L culture conducted in Terrific broth (TB) using affinity chromatography performed using a HiTrap Talon crude column (1 mL, GE Healthcare) and an Äkta start system (GE Healthcare). The protein was eluted using a gradient of imidazole in lysis buffer A (50 mM NaH$_2$PO$_4$, 300 mM NaCl, 10 mM imidazole [pH 8.0]), according to the Äkta system built-in column protocol. The collected proteins were stored at –80°C.

**ChIP-qPCR.** For the detailed protocol, please see the supplemental material. Briefly, to study the binding of AdpA to selected regions of the *S. venezuelae* chromosome, a chromatin immunoprecipitation (ChIP) assay was performed as described in reference 41 with minor changes. For quantification of the DNA fragments, the ChIP was followed by qPCR using the primers listed in Table S2. Sven_WT and Sven_ΔadpA/adpA-FLAG (ΔadpA complementation) strains grown in MYM medium were sampled at 12 and 20 h and subjected to *in vivo* formaldehyde cross-linking. After cell lysis, the protein-DNA complexes

were collected with anti-FLAG M2 magnetic beads (catalog no. M8823, Sigma-Aldrich). The DNA fragments were subsequently purified and used directly in a qPCR assay. The relative quantity (RQ) of each fragment was quantified with the $\Delta\Delta$Ct method, using the *ftsZ* gene as the normalizer and Sven_WT at 12 h as the calibrator strain.

**SDS-PAGE and Western blotting.** For details on sample preparation, SDS-PAGE, and Western blotting conditions, please see the supplemental material.

**Luciferase reporter assay.** The activities of *adpA* promoters in *Streptomyces* hosts were assessed using luciferase reporter assays. Briefly, *Streptomyces* strains containing luciferase reporter plasmids (pFLUXH derivatives; for plasmid and strain construction details, see the supplemental material) were spotted onto the surface of solid DNA or MYM medium (for *S. coelicolor* and *S. venezuelae*, respectively) without antibiotics. The media had been previously placed into the wells (200 $\mu$L per well) of white opaque 96-well plates (OptiPlate-96, Perkin Elmer). Each well was inoculated with 7.5 $\mu$L of activated (10 min, 50°C) spore suspension ($OD_{600}$ = 3.3) prepared in 10% glycerol (for *S. coelicolor*) or 5 $\mu$L of spore suspension ($OD_{600}$ = 2.5) prepared in 10% glycerol (for *S. venezuelae*). Cultures were incubated in the inverted position (lid side down) at 30°C and luminescence readings were collected every 24 or 2 h (for *S. coelicolor* and *S. venezuelae*, respectively) using an EnVision 2105 multimode plate reader (Perkin Elmer). Three clones per strain and four technical replicates per clone were used in each experiment, with the exception of the control strains containing empty pFLUXH plasmids, for which two clones in four technical replicates were tested. The data represent the average of the technical replicate medians.

**Microscopy of *S. venezuelae*.** For microscopic observation of phenotypes, strains were grown in liquid culture for 24 h at 30°C and 250 rpm in 50 mL MYM medium (without antibiotics) in plain flasks without springs. Samples were collected at 2-h intervals, transferred to glass slides, air-dried, layered with 10% glycerol in phosphate-buffered saline (PBS), and covered with a coverslip. Images were acquired under the 100× DIC HC PL Apo Objective of a Leica DM6 B microscope using a Leica DFC7000 GT camera and manipulated with the LAS X software.

**Disk diffusion assays to assess antibacterial activity.** The method is described in detail in the supplemental material. In brief, to assess chloramphenicol production in liquid cultures, the modified Kirby-Bauer disk diffusion method (87) was applied. Sterile paper disks were soaked with concentrated culture supernatant extracts of respective *S. venezuelae* strains and tested for antibiotic activities using indicator strains. To estimate chloramphenicol production levels, a standard curve of growth inhibition zones was generated using disks soaked with solutions containing defined amounts of commercially available antibiotic, as tested against a lawn culture of *M. luteus*.

**Electrophoretic mobility shift assay.** EMSA is described in detail in in the supplemental material. Briefly, EMSA was performed using Cy5 fluorescently labeled DNA fragments and purified recombinant AdpASv_His and AdpASc_His proteins. The reactions were performed in HBS200 buffer supplemented with bovine serum albumin (BSA) and poly(dI-dC)·poly(dI-dC) competitors, and the samples were resolved in 5% native polyacrylamide gels as described previously (33). The gels were analyzed using a ChemiDocMP system (Bio-Rad).

**RT-qPCR.** For details on RNA isolation and RT-qPCR analysis, please see the supplemental material.

**Bioinformatics analyses.** The multiple sequence alignments of amino acids were performed with ClustalW (parameters: gap opening penalty, 15,66; gap extension penalty, 6.66; weight matrix, ClustalW; iteration type, alignment) using the Ugene software (88). See also the supplemental material.

**Nucleotide and amino acid sequences.** The genome sequence of *S. venezuelae* strain NRRL B-65442 (WT) is available from the NCBI under reference accession numbers CP018074.1 and CP018075.1 (75). The nucleotide and amino acid sequences of *adpA_Sv* and AdpASv, respectively, from *S. venezuelae* are available from http://streptomyces.org.uk/ under number vnz_12630.

## SUPPLEMENTAL MATERIAL

Supplemental material is available online only.

**SUPPLEMENTAL FILE 1**, PDF file, 0.4 MB.
**SUPPLEMENTAL FILE 2**, PDF file, 2.9 MB.

## ACKNOWLEDGMENTS

M.P., J.Z.-C., and M.W. designed the experiments. M.P., M.K., and M.W. performed the research. M.P., J.Z.-C., and M.W. wrote the manuscript. We thank Abhishek Kumar Singh and Olimpia Żuchowicz (University of Wrocław) for their help with the preparation of disk diffusion assays and protein purification.

This study was funded by the National Science Centre (Poland) MAESTRO grant 2012/04/A/NZ1/00057 to J.Z.-C. and Miniatura grant 2019/03/X/NZ1/00280 to M.W. The cost of publication was financed by the Excellence Initiative - Research University (IDUB) program for the University of Wrocław and by the Faculty of Biotechnology statutory funds.

We declare that the research was conducted in the absence of any commercial or financial relationships that could be construed as a potential conflict of interest. We declare no conflicts of interest.

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
