## [Reviewer comments · Microbiology Spectrum]

Microbiology Spectrum

AdpA positively regulates morphological differentiation and chloramphenicol biosynthesis in *Streptomyces venezuelae*

Małgorzata Płachetka, Michał Krawiec, Jolanta Zakrzewska-Czerwińska, and Marcin Wolański

Corresponding Author(s): Marcin Wolański, University of Wrocław

Review Timeline:

Submission Date:

October 22, 2021

Accepted:

October 29, 2021

Editor: Jeffrey Gralnick

Reviewer(s): The reviewers have opted to remain anonymous.

Transaction Report:

DOI: <https://doi.org/10.1128/Spectrum.01981-21>

October 29, 2021

Dr. Marcin Przemysław Wolański
University of Wrocław
Biotechnology
Joliot-Curie 14A
Wrocław 50-383
Poland

Re: Spectrum01981-21 (AdpA positively regulates morphological differentiation and chloramphenicol biosynthesis in *Streptomyces venezuelae*)

Dear Dr. Marcin Przemysław Wolański:

Based on your thoughtful responses and revisions to the prior round of review at JBact, your manuscript has been accepted, and I am forwarding it to the ASM Journals Department for publication. You will be notified when your proofs are ready to be viewed.

Sincerely,

Jeffrey Gralnick
Editor, Microbiology Spectrum

Journals Department
Supplemental file 10: Accept
Supplemental file 1: Accept
Supplemental file 6: Accept
Supplemental file 3: Accept
Supplemental file 4: Accept
Supplemental file 7: Accept
Supplemental file 9: Accept
Supplemental file 2: Accept
Supplemental file 8: Accept
Supplemental file 5: Accept